# The Association Between Chronic Obstructive Pulmonary Disease (COPD) and Migraine: Systematic Review and Meta-Analysis

**DOI:** 10.3390/jcm13226944

**Published:** 2024-11-18

**Authors:** Saleem Alshehri, Maha Saad Zain Al-Abeden, Mona Aldukain, Ali Aldukain, Dhai Almuteri, Assal Hobani, Abdulmalik Barakat, Nora Alzoum

**Affiliations:** 1Faculty of Medicine, King Khalid University, Abha 62521, Saudi Arabia; s.alshehry@hotmail.com (S.A.); mona.aldukain11@gmail.com (M.A.); dr.alihq@gmail.com (A.A.); 2College of Medicine, Sulaiman Al Rajhi University, Al-Bukairiyah 52726, Saudi Arabia; 3Faculty of Medicine, Qassim University, Unaizah 51911, Saudi Arabia; dhayzabar1@gmail.com; 4Faculty of Medicine, Ibn Sina National College for Medical Studies, Jeddah 22421, Saudi Arabia; assal7obani@gmail.com; 5Faculty of Medicine, King Abdulaziz University, Jeddah 21589, Saudi Arabia; abdulmalikbarakat3@gmail.com; 6College of Medicine, Princess Nourah bint Abdulrahaman University, Riyadh 11671, Saudi Arabia; 439001498@pnu.edu.sa

**Keywords:** COPD, migraine, comorbidity, systematic inflammation, risk factors, chronic illness

## Abstract

**Background/Objectives**: Migraine and chronic obstructive pulmonary disease (COPD) are both common chronic conditions that may share underlying pathophysiological mechanisms despite presenting with distinct clinical features. Understanding the association between these two conditions could enhance treatment strategies and improve patient outcomes. This review aims to evaluate the relationship between COPD and migraine, highlighting shared risk factors and identifying gaps in the existing literature. **Methods**: A search was conducted across four electronic databases (PubMed, Ovid Medline, ScienceDirect, and Web of Science) following PRISMA guidelines up to January 2024. The search identified 85 studies, of which five met the inclusion criteria: observational studies, cohort studies, case–control studies, and randomized controlled trials examining the association between migraine and COPD. Studies not published in English and irrelevant study designs were excluded. The risk of bias was assessed using the JBI Critical Appraisal Tool, which evaluated aspects such as study design, participant selection, measurement methods, and the handling of confounding factors. **Results**: The review included five studies comprising 184,817 patients. All studies identified a significant association between COPD and migraine, with varying methodologies for diagnosing the conditions. Notably, COPD patients had an increased risk of migraine compared to controls, and migraine patients also demonstrated a higher risk of developing COPD. However, the evidence was marked by high heterogeneity and potential confounding factors. **Conclusions**: The findings suggest a significant association between COPD and migraine, potentially driven by shared mechanisms such as systemic inflammation. However, the predominance of cross-sectional studies limits causal inference. Future research should prioritize longitudinal studies to clarify the directionality and causality of the relationship between COPD and migraine while thoroughly addressing potential confounding factors.

## 1. Introduction

Chronic obstructive pulmonary disease (COPD) is a major contributor to global morbidity and mortality, posing a significant socio-economic challenge. This chronic inflammatory condition is marked by airflow obstruction during breathing. The World Health Organization (WHO) reported that COPD was the third leading cause of death globally in 2019, accounting for 3.23 million deaths [1]. It is the third most common noncommunicable disease. It is further predicted to increase in the upcoming decades because of continued exposure to COPD risk factors and the ageing of the overall population [2,3]. Defining the daily burden of COPD consists of several symptoms that have a great impact. The most common are dyspnea or difficulty breathing, cough, and sputum production [4].

Headache disorders affect approximately 40% of the global population, equating to about 3.1 billion people in 2021. Although there are regional differences, headache disorders are a pervasive issue, impacting individuals across all races, income levels, and geographic locations. According to the Global Health Estimates 2019, headache disorders were identified as the third leading cause of disability-adjusted life years (DALYs) worldwide, following stroke and dementia [5]. Migraine is a severe neurobiological headache disorder marked by a combination of neurological, gastrointestinal, and autonomic disturbances. Common symptoms include sudden, unilateral headaches, nausea, vomiting, and heightened sensitivity to light (photophobia) and sound (phonophobia), which can significantly impair overall quality of life. Additionally, 20–60% of individuals with migraines experience premonitory symptoms hours to days before an attack, such as depression and cognitive dysfunction [6].

Both migraine and chronic obstructive pulmonary diseases (COPDs) are recognized as common chronic comorbid medical conditions that have episodic exacerbations and significant illness-related disability. Many studies have sought to highlight the association between migraine and respiratory illness through epidemiological studies. As early as 1977 and 1988, asthma has been described as “acephalic migraine” and “pulmonary migraine” [7]. Several years later, further cross-sectional studies evidenced such a relationship whereby asthma has been found to be 1.1–1.6 times more prevalent in persons with migraine [8,9]. The study by Aamodt et al. [9] found a strong association between the prevalence of asthma and self-reported frequency of migraine attacks with an odds ratio of 1.6, 2.2, and 2.9 for those in the low, moderate, and high-frequency categories. Moreover, another recent study in 2019 by Kim et al. highlighted that both asthma and migraines are reciprocally associated, whereby the asthma group has demonstrated an adjusted hazard ratio of 1.47 for migraine (95% CI = 1.41–1.53, *p* < 0.001). While the migraine group has shown a similar adjusted hazard ratio of 1.37 for asthma (95% CI = 1.33–1.41, *p* < 0.001) [10].

Similarly, only a few recent studies have assessed the relationship between chronic hypoxemia, which results from COPD, and headaches. COPD is characterized by a limitation of airflow that is not fully reversible; it continues to progress further, leading to hypoxemia and hypercarbia, which may have a pathophysiological role in the development of headaches. A study by Ozge et al. [11] showed that 31.9% of the patients diagnosed with COPD have complained of headaches. Such a literature review has caught our attention in suggesting a potential link between migraine and COPD that warrants further investigation. Therefore, this systematic review and meta-analysis aimed to explore the potential association between these comorbid conditions.

## 2. Materials and Methods

We performed a systematic review and meta-analysis in accordance with the Preferred Reporting Items for Systematic Reviews and Meta-Analyses (PRISMA) guidelines (version 5.1.0) [12,13], following the Cochrane Handbook for Systematic Reviews of Interventions [14]. The study protocol was pre-registered in the International Prospective Register of Systematic Reviews (PROSPERO) (ID: CRD42024513222).

### 2.1. Literature Search and Data Collection

Our search strategy involved electronic databases such as PubMed, Ovid Medline, ScienceDirect, and Web of Science. Below is a summary (Table 1) of the search strategies employed for each database. The predefined search terms were (“Chronic Obstructive Lung Disease” OR “COAD” OR “COPD” OR “Chronic Obstructive Airway Disease” OR “Chronic Obstructive Pulmonary Disease” OR “Chronic Airflow Obstruction”) AND (“Migraine Disorder” OR “Migraine” OR “Migraines” OR “Migraine Headache” OR “Migraine Headaches”).

### 2.2. Eligibility Criteria and Study Selection

Three independent reviewers conducted the initial screening based on predefined inclusion and exclusion criteria. Clinical trials, observational studies, and case series focusing on the association between migraine and chronic obstructive pulmonary disease without time restriction were considered eligible. On the other hand, non-English studies and review articles were excluded.

### 2.3. Data Extraction

Data extraction was carried out by two independent reviewers using a standardized format to record key study information, including author, year, title, and journal. Details captured included study design, participant demographics (gender, age, and total number of participants), specifics of chronic obstructive pulmonary disease (diagnosis criteria, definition, severity, treatment, and comorbidities), and migraine-related information (diagnosis criteria, definition, prevalence, incidence, and comorbidities). Outcome measures related to both chronic obstructive pulmonary disease and migraine were also documented.

### 2.4. Quality Assessment

The methodological quality of the included studies was assessed using the JBI Critical Appraisal Tool [15]. Each study was independently evaluated by two reviewers for potential biases, participant selection, study design, and methodology. Any discrepancies were resolved through consensus between the reviewers or by consulting a third reviewer.

### 2.5. Statistical Analysis

A random-effects meta-analysis was conducted using RevMan (Version 5.3) and Open meta-analysis for Microsoft Windows. Our primary effect size measure was the odds ratio (OR) with its 95% confidence interval (CI). The odds ratios included two different models. In the first model, migraine was considered the dependent variable, and COPD was considered the independent variable. This model examines whether the presence of COPD increases the risk of migraine (i.e., the risk of migraine among COPD patients). In the second model, migraine was considered the independent variable, and COPD was considered the dependent variable. This model examines whether migraine patients exhibit an increased risk of COPD (i.e., the risk of COPD among migraine patients). A *p*-value < 0.05 indicated a statistically significant effect size. To assess heterogeneity, the I^2^ statistic was used. An I^2^ value of 50% with a *p*-value < 0.1 was considered indicative of statistically significant heterogeneity.

## 3. Results

### 3.1. Search Results

The PRISMA flowchart is presented in Figure 1. The initial search identified 85 records, with nine duplicates removed prior to screening. Title and abstract screening excluded 69 articles, leaving seven for full-text evaluation. During the full-text review, two of these seven articles were excluded for not meeting the inclusion criteria: one for reporting the wrong outcome (n = 1) and another for involving the wrong population (n = 1). Ultimately, four cross-sectional [16,17,18,19] and one case–control [20] study were selected for further evaluation.

### 3.2. The Characteristics of the Included Studies

Table 2 and Table 3 present the details of the included studies. These studies were published between 2002 and 2019 and included 184,817 patients. For the diagnosis of migraine, two studies used standardized questionnaires [17,18], and one study used the International Classification of Headache Disorders-2 (ICHD-2) criteria [16]. Additionally, Davey et al. (2002) used a combination of diagnostic and medication codes to diagnose migraine patients [20]. COPD diagnosis criteria varied as well: two studies made the diagnosis through reported symptoms or by being told by a healthcare professional that they had emphysema or chronic bronchitis in the last 12 months [17,19]. One study defined COPD based on OXIMIS diagnostic codes [20]. The remaining two studies did not mention the criteria for diagnosing COPD [16,18].

The included studies report significant associations between migraine and COPD. Miguel-Díez et al. (2018) found that the prevalence of migraine was higher among COPD patients (22.5%) compared to controls (13.2%), suggesting a higher risk of migraine among COPD patients [19]. Similarly, Minen et al. (2019) reported a significant association, attributing this relationship possibly to sleep disturbances or airway constriction caused by COPD. In examining the risk of COPD among migraine patients [17], Buse et al. (2010) found that COPD prevalence was higher in chronic migraine sufferers (4.89%) compared to those with episodic migraine (2.6%) [16]. Davey et al. (2002) also reported a higher relative risk (RR) of COPD in migraine patients compared to those without migraine [20]. Lastly, Wang et al. (2016) observed a significantly higher prevalence of COPD among migraine patients (8.8%) versus non-migraine patients (2.3%) [18].

### 3.3. Risk of Bias Assessment

Based on the JBI Critical Appraisal Checklists for Case-Control and Cross-Sectional Studies, the included studies exhibit high methodological quality, as illustrated in Appendix A. Each study carefully addressed design, participant selection, exposure measurement, outcome assessment, and confounding factors. This meticulous approach enhances the internal validity and reliability of the findings, providing solid evidence of the association between migraine and COPD.

### 3.4. The Relation Between COPD and Migraine

The prevalence of COPD among migraine patients was found to be 2.3% (95% CI, 1.3%–3.3%) (Figure 2). Our analysis revealed that patients with migraine had a statistically significant increased risk of COPD compared to controls (pooled OR, 1.75; 95% CI, 1.06–2.86), with substantial heterogeneity detected across the studies (I^2^ = 81%) (Figure 3). Similarly, we found that COPD patients had a significantly increased risk of migraine compared to controls (pooled OR, 2.55; 95% CI, 2.38–2.74), although the overall heterogeneity was high (I^2^ = 98%) (Figure 4)

## 4. Discussion

Chronic obstructive pulmonary disease (COPD) and migraine are both common worldwide diseases that significantly impact the quality of life of those affected. COPD is a treatable disease characterized by persistent airflow limitations in the lungs, often caused by exposure to harmful substances [21]. Conversely, migraine is a neurological disorder that involves recurring headaches, often with associated nausea and increased sensitivity to light and sound [22].

This systematic review aimed to explore the potential association between COPD and migraine. Early evidence, such as a 2002 case–control study, suggested an increased risk of COPD among migraine patients [20]. More recent studies, like a cross-sectional study, found a notable prevalence of allergic and respiratory disorders, including COPD, among migraine patients [23]. A Spanish study comparing adults with COPD and matched controls reported a higher migraine prevalence in COPD patients (22.5%) compared to non-COPD individuals (13.2%) [19]. In 2019, Minen et al. used data from the 2013–2015 national health survey and identified a twofold increase in the association between migraine or severe headaches and COPD [17]. These findings may suggest that shared risk factors such as inflammatory pathways, oxidative stress, and other biological processes may underlie the connection between COPD and migraine.

Miguel-Díez et al. (2018) found a significantly higher prevalence of migraine in COPD patients compared to controls, highlighting important considerations for COPD management [19]. Similarly, Minen et al. (2019) demonstrated a 15.1% prevalence of migraine in COPD patients, attributing this link to factors like sleep disturbances and airway constriction related to COPD [17]. Conversely, Buse et al. (2010) found that chronic migraine patients had a higher risk of developing COPD than those with episodic migraine, with a stronger association in women [16].

Davey et al. (2002) conducted a large case–control study, revealing an elevated relative risk of COPD in migraine patients, supporting the hypothesis of shared mechanisms, such as chronic inflammation, contributing to both conditions [20]. Similarly, Wang et al. (2016) reported a significantly higher prevalence and increased risk of COPD among migraine patients [18].

The relationship between COPD and migraine may be explained by shared inflammatory pathways, oxidative stress, and other factors influenced by smoking and systemic inflammation [22,24]. COPD is characterized by chronic airway inflammation and impaired lung function due to harmful substance exposure, such as cigarette smoking, which may affect neurological processes linked to migraine [24]. Conversely, migraine, characterized by neurogenic inflammation and vascular changes [25], may be worsened by the systemic inflammation and oxidative stress seen in COPD. Moreover, sleep disturbances are common among COPD patients [26], which may contribute to worsening migraine symptoms, suggesting a possible relationship between the two conditions.

Our systematic review identified several limitations in the existing literature. First, only five studies were available that assessed the association between COPD and migraine, which limits the robustness and generalizability of the findings. Second, unmeasured confounding factors, such as lifestyle variables (e.g., smoking) and comorbid conditions (e.g., anxiety), may have influenced the observed associations. Lastly, most of the included studies were cross-sectional, which restricts the ability to establish causality. Therefore, this systematic review and meta-analysis aimed to explore the potential association between COPD and migraine, acknowledging that while shared risk factors may contribute to both conditions, causality cannot be definitively established due to the predominance of cross-sectional and case–control studies in the current literature.

Based on our review, several recommendations for future research are proposed. The limited number of studies on this association underscores the need for more comprehensive research. Conducting more longitudinal studies would provide a better understanding of the causal relationship between COPD and migraine. Future studies should also consider potential confounding factors to yield more accurate results. Our findings suggest an observed association between COPD and migraine. Educating patients about the potential relationship between these conditions could help raise awareness and support more informed management. Promoting lifestyle modifications, such as smoking cessation, improving sleep quality, and maintaining a healthy diet, may also help in managing both conditions more effectively. However, these strategies should be considered as part of a holistic care plan, pending further research to better understand the nature of the association between COPD and migraine.

## 5. Conclusions

The review findings revealed a significant association between COPD and migraine, indicating a bidirectional relationship. Patients with migraine were found to have a higher risk of developing COPD, and conversely, those with COPD were at an increased risk of experiencing migraines compared to control groups.

These results underscore the importance of recognizing the comorbidity between migraine and COPD. This insight can guide future research, inform clinical best practices, and foster the development of integrated management strategies for patients affected by both conditions. Understanding the connection between COPD and migraine is crucial for enhancing comprehensive care and improving treatment outcomes for these individuals.

## Figures and Tables

**Figure 1 jcm-13-06944-f001:**
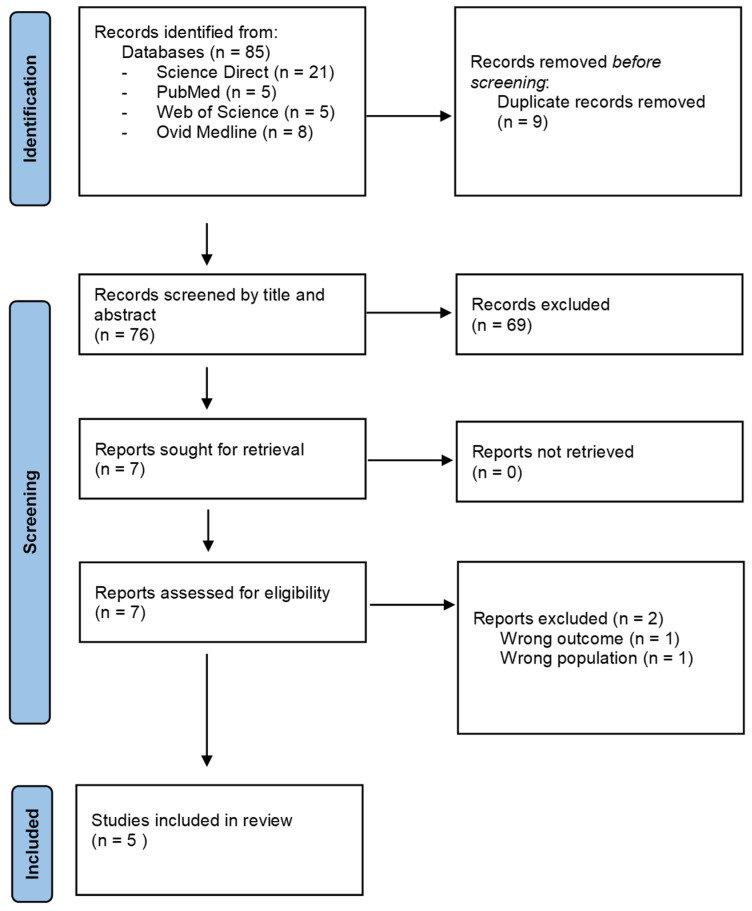
PRISMA flow diagram.

**Figure 2 jcm-13-06944-f002:**
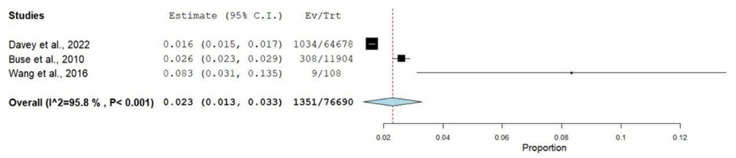
Forest plot showing the prevalence of COPD among migraine patients [16,18,20]. The black square symbolizes the effect size of each study. The blue diamond is the pooled effect size. The red dotted line is the reference line.

**Figure 3 jcm-13-06944-f003:**
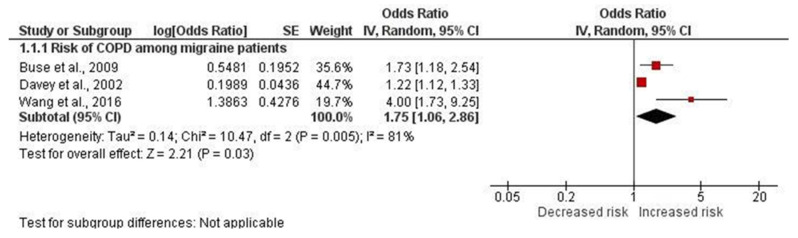
Forest plot showing the odds of COPD in patients with migraine [16,18,20]. The red square is the effect size of each study. The black diamond is the pooled effect size.

**Figure 4 jcm-13-06944-f004:**
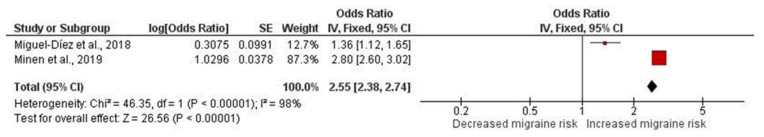
Forest plot showing the odds of migraine in patients with COPD [17,19]. The red square is the effect size of each study. The black diamond is the pooled effect size.

**Table 1 jcm-13-06944-t001:** Search strategies for each database.

Database	Search Terms	Search Date
PUBMED	(“Chronic Obstructive Lung Disease” [Title/Abstract] OR “COAD” [Title/Abstract] OR “COPD” [Title/Abstract] OR “Chronic Obstructive Airway Disease” [Title/Abstract] OR “Chronic Obstructive Pulmonary Disease” [Title/Abstract] OR “Chronic Airflow Obstruction” [Title/Abstract]) AND (“Migraine Disorder” [Title/Abstract] OR “Migraine” [Title/Abstract] OR “Migraines” [Title/Abstract] OR “Migraine Headache” [Title/Abstract] OR “Migraine Headaches” [Title/Abstract])	up to January 2024
OVID MEDLINE	(“Chronic Obstructive Lung Disease” OR “COAD” OR “COPD” OR “Chronic Obstructive Airway Disease” OR “Chronic Obstructive Pulmonary Disease” OR “Chronic Airflow Obstruction”) AND (“Migraine Disorder” OR “Migraine” OR “Migraines” OR “Migraine Headache” OR “Migraine Headaches”)	up to January 2024
ScienceDirect	(“Chronic Obstructive Lung Disease” OR “COAD” OR “COPD” OR “Chronic Obstructive Airway Disease” OR “Chronic Obstructive Pulmonary Disease” OR “Chronic Airflow Obstruction”) AND (“Migraine Disorder” OR “Migraine” OR “Migraines” OR “Migraine Headache” OR “Migraine Headaches”)	up to January 2024
WEB OF SCIENCE	((AB = (“Chronic Obstructive Lung Disease” OR “COAD” OR “COPD” OR “Chronic Obstructive Airway Disease” OR “Chronic Obstructive Pulmonary Disease” OR “Chronic Airflow Obstruction”)) AND AB = (“Migraine Disorder” OR “Migraine” OR “Migraines” OR “Migraine Headache” OR “Migraine Headaches”))	up to January 2024

**Table 2 jcm-13-06944-t002:** Characteristics of the included studies.

Study ID	Study Design	Sample Size	Study Groups	Age Range	Gender Males, n (%)	Definition or Diagnosis Criteria for COPD	Definition or Diagnosis Criteria for Migraine
Risk of migraine among COPD patients							
Miguel-Díez et al., 2018 [19]	Cross-sectional	804	Migraine in COPD vs. no COPD	35 to 80 +	188 (23.3)	Symptoms: dyspnea, cough, and sputum production	NR
Minen et al., 2019 [17]	Cross-sectional	104,843	Migraine in COPD vs. no COPD	18 to 65+	50,534 (48.2)	Having ever been told by a health professional that they had emphysema or having been told in the past 12 months that they had chronic bronchitis.	NHIS survey was made by asking participants whether they had experienced migraine or severe headache in the past three months
Risk of COPD among migraine patients							
Buse et al., 2010 [16]	Cross-sectional	11,904	COPD in chronic vs. episodic migraine	18+	2920 (24.5)	NR	Chronic migraine (ICHD-2 defined migraine; ≥15 days of headache per month) and episodic migraine (ICHD-2 defined migraine; 0–14 days of headache per month)
Davey et al., 2002 [20]	Case-control	64,678	COPD in migraine vs. no migraine	NR	16,107 (24.9)	COPD’s definition was defined according to combinations of OXMIS codes that have been developed and validated earlier.	A combination of OXMIS diagnostic codes and British National Formulary medication codes
Wang et al., 2016 [18]	Cross-sectional	2588	COPD in migraine vs. no migraine	18 to 75+	435 (38)	NR	Chinese version of ID-Migraine

COPD: Chronic obstructive pulmonary disease; NR: Not reported; NA: Not applicable.

**Table 3 jcm-13-06944-t003:** Summary of studies’ outcomes.

Study ID	Primary Outcome (s)	Prevalence of Migraine Among COPD	COPD Prevalence Among Migraine	Main Finding	Conclusion
Risk of migraine among COPD patients					
Miguel-Díez et al., 2018 [19]	Risk of migraine in COPD patients.	COPD vs. Controls: 22.5% vs. 13.2%	NA	Risk of migraine among COPD patients: OR: 1.36, 95% CI [1.12, 1.65]	The prevalence of chronic neck pain, chronic low back pain and migraine was significantly higher among COPD patients in comparison with controls having none of these diseases.
Minen et al., 2019 [17]	Association between migraine and COPD	0.151	NA	Risk of migraine among COPD patients: OR: 2.80, 95% CI [2.60, 3.02]	A statistically significant association was found between migraine or severe headache and COPD. This relationship may be attributed to sleep disturbances associated with headaches or airway constriction linked to COPD.
Risk of COPD among migraine patients					
Buse et al., 2010 [16]	Sociodemographic profiles and the frequency of comorbidities for adults with chronic migraine and episodic migraine.	NA	4.89% in chronic, 2.6% in episodic	Risk of COPD among migraine patients: OR: 1.73, 95% CI [1.18, 2.54]	Individuals with chronic migraine were significantly more likely to report respiratory disorders, such as asthma, bronchitis, and COPD, as well as cardiac risk factors, including hypertension, diabetes, high cholesterol, and obesity, compared to those with episodic migraine.
Davey et al., 2002 [20]	Risk of COPD in migraine patients.	NA	1.6% in migraine, 1.3% in no migraine	Risk of COPD among migraine patients: RR: 1.22, 95% CI [1.12, 1.33]	The relative risk of COPD was significantly higher among patients with a history of migraine than among those without a history of migraine.
Wang et al., 2016 [18]	Validation of the Chinese version of the ID-Migraine	NA	8.8% in migraine, 2.3% in no migraine	Risk of COPD among migraine patients: OR: 4.00, 95% CI [1.73, 9.25]	Age, sex, education level, depression, CHD, COPD, and hypertension were significantly associated with migraine.

COPD: Chronic obstructive pulmonary disease; NR: Not reported; NA: Not applicable; OR: Odds ratio; RR: Risk ratio.

## Data Availability

The data supporting the findings of this meta-analysis are publicly available in the included studies, as cited in the manuscript.

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
