# Peer review of "The Association Between Chronic Obstructive Pulmonary Disease (COPD) and Migraine: Systematic Review and Meta-Analysis"

_jcm, 2024, doi:10.3390/jcm13226944_

Round 1

Reviewer 1 Report

Comments and Suggestions for Authors

Comments to authors

Introduction

At the end of the section, you should describe exactly what your review is about, not just say that a review is needed. It could be "Therefore, this systematic review and meta-analysis aimed to evaluate ........".

Materials and Methods

Lines 93-96: Cite MOOSE in addition to PRISMA, as MOOSE is a specific checklist for systematic reviews of observational studies.

The methodology should not describe the results of the search (number of studies found, how many were included, etc.). This should be in the first section of the results.

Eligibility Criteria and Study Selection: Missing exclusion criteria.

The methodology (statistical analysis) should describe the type of meta-analysis performed (" random " or " fixed " model). The reference to the type of meta-analysis performed should also be given, as well as the citation for the assessment of heterogeneity (the Cochrane Handbook is usually used).

Results

There are sections of the manuscript that are not justified (alignment).

Discussion

Be more cautious in interpreting observed associations. If there are no longitudinal studies (cohorts) to support causality, but only cross-sectional or case-control studies, it is reasonable to consider COPD as a risk factor for migraine. At most, there may be common risk factors for both conditions. But it is difficult to justify that migraine causes COPD. In reality, with this type of study (cross-sectional and similar), the authors can only establish an association between A and B, but they cannot establish causality. Only by using other causality criteria, such as biological plausibility, temporal precession, etc., can they justify that A causes B. And if you only include association studies in your review, and take biological plausibility into account, you can only hypothesise that COPD worsens or causes migraine.

References

Some references are written incorrectly (only the authors' initials appear, etc.).

Author Response

Comment 1: Introduction

At the end of the section, you should describe exactly what your review is about, not just say that a review is needed. It could be "Therefore, this systematic review and meta-analysis aimed to evaluate ........".

Response 1: Thank you for pointing this out. We agree with this comment. Therefore, we have well-clarified our aim in the revised manuscript. (Line 95-96)

Comment 2: Materials and Methods

Lines 93-96: Cite MOOSE in addition to PRISMA, as MOOSE is a specific checklist for systematic reviews of observational studies.

 Response 2: Thank you for the relevant suggestion. Three citations has been added numbered as 12, 13 and 14 in the references.

The methodology should not describe the results of the search (number of studies found, how many were included, etc.). This should be in the first section of the results.

  Response 3: Thank you for pointing this out. The search results have been shifted from (Line 111-112) in the methodology section and revised in the first paragraph of results section (Line 156-158).

Eligibility Criteria and Study Selection: Missing exclusion criteria.

Response 4: Thank you for the meticulous reading and critical appraisal. We have added detail of the inclusion and exclusion criteria in the section for better clarity, line 123-124.

The methodology (statistical analysis) should describe the type of meta-analysis performed (" random " or " fixed " model). The reference to the type of meta-analysis performed should also be given, as well as the citation for the assessment of heterogeneity (the Cochrane Handbook is usually used).

Response 5: We thank the reviewer for the useful suggestion to add critical thoughts. We have added the meta-analysis model; lines 142-143. The assessment of heterogeneity followed the Cochrane Handbook, and was accordingly cited; line 102.

Comment 3: Results

There are sections of the manuscript that are not justified (alignment).

Response 6: Thank you for the meticulous reading and attention to details, the section has been modified accordingly.

Comment 4: Discussion

Be more cautious in interpreting observed associations. If there are no longitudinal studies (cohorts) to support causality, but only cross-sectional or case-control studies, it is reasonable to consider COPD as a risk factor for migraine. At most, there may be common risk factors for both conditions. But it is difficult to justify that migraine causes COPD. In reality, with this type of study (cross-sectional and similar), the authors can only establish an association between A and B, but they cannot establish causality. Only by using other causality criteria, such as biological plausibility, temporal precession, etc., can they justify that A causes B. And if you only include association studies in your review, and take biological plausibility into account, you can only hypothesise that COPD worsens or causes migraine.

Response 7: We thank the reviewer for the useful suggestion to add critical thoughts. We agree that providing our thoughts and critical appraisal of the current state of the art based on the limitations would add value to our review. We have addressed that point by thoroughly revising the discussion section, kindly refer to lines 271-278.

Comment 5: References

Response 8: Thank you for the meticulous reading and attention to details, the section has been modified accordingly.

Reviewer 2 Report

Comments and Suggestions for Authors

Alshehri et al. have submitted a comprehensive systematic review and meta-analysis that investigates the association between COPD and migraine. The paper is well-written, with a clear and well-defined objective, and offers valuable insights into the relationship between these two chronic conditions. However, there are areas that could benefit from improvement to enhance the clarity and impact of the review.

The Introduction is well presented and sets the stage for the review by providing relevant background information and highlighting the importance of the research question. It successfully contextualizes the study and underscores the significance of exploring the link between COPD and migraine.

In the Materials and Methods section, particularly Section 2.2 (Eligibility Criteria and Study Selection), the authors have mentioned that only five studies met the inclusion criteria following full-text assessment, However, the specific inclusion and exclusion criteria are not detailed. For clarity and transparency, I recommend that the authors provide a more comprehensive explanation of these criteria. This addition would help readers understand the basis for the selection of studies and improve the replicability of the review.

The results section is thorough, and the statistical analysis, particularly the meta-analysis, is well-executed and presented clearly. I have no significant concerns regarding this section.

In the last paragraph of the Discussion section, the authors recommend more comprehensive management strategies for COPD patients based on their findings. I take issue with this strong recommendation because the evidence, although statistically significant, is limited by the constraints outlined by the authors themselves. The predominance of cross-sectional studies among the included research, for example, limits the ability to draw causal inferences. I suggest revising this section to better reflect these limitations and propose that future research is needed to support any changes in clinical management.

Author Response

Comments and Suggestions for Authors

Alshehri et al. have submitted a comprehensive systematic review and meta-analysis that investigates the association between COPD and migraine. The paper is well-written, with a clear and well-defined objective, and offers valuable insights into the relationship between these two chronic conditions. However, there are areas that could benefit from improvement to enhance the clarity and impact of the review.

Comment 1: The Introduction is well presented and sets the stage for the review by providing relevant background information and highlighting the importance of the research question. It successfully contextualizes the study and underscores the significance of exploring the link between COPD and migraine.

Response 1: We would like to express our appreciation and gratitude for your cheerful comments. We are proud that our paper met your expectations to be beneficial in the field.

Comment 2: In the Materials and Methods section, particularly Section 2.2 (Eligibility Criteria and Study Selection), the authors have mentioned that only five studies met the inclusion criteria following full-text assessment, However, the specific inclusion and exclusion criteria are not detailed. For clarity and transparency, I recommend that the authors provide a more comprehensive explanation of these criteria. This addition would help readers understand the basis for the selection of studies and improve the replicability of the review.

Response 2: Thank you for the meticulous reading and critical appraisal. We have added detail of the inclusion and exclusion criteria in the section for better clarity; line 123-124.

Comment 3: The results section is thorough, and the statistical analysis, particularly the meta-analysis, is well-executed and presented clearly. I have no significant concerns regarding this section.

Response 3: We would like to thank you for your appreciation of our work and are happy to see that it met your liking.

Comment 4: In the last paragraph of the Discussion section, the authors recommend more comprehensive management strategies for COPD patients based on their findings. I take issue with this strong recommendation because the evidence, although statistically significant, is limited by the constraints outlined by the authors themselves. The predominance of cross-sectional studies among the included research, for example, limits the ability to draw causal inferences. I suggest revising this section to better reflect these limitations and propose that future research is needed to support any changes in clinical management.

Response 4: We thank the reviewer for the useful suggestion to add critical thoughts. We agree that providing our thoughts and critical appraisal of the current state of the art based on the limitations would add value to our review. We have addressed that point by thoroughly revising the discussion section, kindly refer to lines 271-278.
